# Applications of Growth Factors in Implant Dentistry

**DOI:** 10.3390/cimb47050317

**Published:** 2025-04-28

**Authors:** Balen Hamid Qadir, Mohammed Aso Abdulghafor, Mohammed Khalid Mahmood, Faraedon Mostafa Zardawi, Mohammed Taib Fatih, Handren Ameer Kurda, Zana Fuad Noori, Herve Tassery, Delphine Tardivo, Arthur Falguiere, Vincent Romao, Romain Lan

**Affiliations:** 1Department of Dentistry, Komar University of Science and Technology, Sulaimani 46001, Kurdistan, Iraq; 2College of Dentistry, Sulaymaniyah University, Sulaimani 46001, Kurdistan, Iraq; 3Faculty of Medical and Paramedical Sciences, Aix-Marseille University, French National Center of Scientific Research (CNRS), French Blood Establishment (EFS), Bio-Cultural Anthropology, Law, Ethics and Health Laboratory (ADES), 13005 Marseille, France; 4Dean of the Faculty of Dentistry, Qaiwan International University, Sulaimani 46001, Kurdistan, Iraq; 5Department of Dentistry, American University of Sulaimani Iraq AUIS, Sulaimani 46001, Kurdistan, Iraq; 6Dental School of Medicine, Conservative and Endodontic Department, Aix-Marseille University, 13005 Marseille, France; 7Marseille Hospital APHM, IHU-MEPHI Institute, 13005 Marseille, France; 8Oral Surgery Department, Timone University Hospital, 264 rue Saint Pierre, 13385 Marseille Cedex 5, France

**Keywords:** growth factors, concentrated growth factors, platelet-derived growth factors, dental implants, osseointegration, regenerative dentistry, wound healing

## Abstract

Growth factors are proteins that play an essential part in tissue regeneration and development. They attach surface receptors to mediate their actions on cells. Signaling systems within cells are activated when growth factors bind to their associated receptors. These signaling cascades control the transcription of genes involved in cellular functions like proliferation, differentiation, migration, protein synthesis, and metabolism. This narrative review provides a comprehensive update on the use of growth factors in implant dentistry with a special emphasis on human clinical trials. Since wound healing and osseointegration are pre-requisites of successful implantation and growth factors are important components of homeostasis and wound healing, this review first starts with the basic biology of wound healing. Then, it presents the specific role of growth factors in wound healing and tissue regeneration. Finally, the PubMed database was searched using relevant keywords with some filters related to the research question. Out of the initial 44 records, all the clinical human studies (*n* = 29) with the actual dental implant placement and its assessment were included. These results of the published and relevant literature over the last 25 years on different applications of growth factors in the field of implant dentistry are critically discussed.

## 1. Introduction

Dental implants aiming to restore normal function and esthetics of missing teeth are widely employed. Under normal circumstances, most methods are scientifically proven and predictable. However, the desired implant site is often not suitable due to poor bone quality or inadequate bone. The implant site’s closeness to the maxillary sinus or mandibular canal can also cause insufficient alveolar ridge height [1,2].

To address these needs, dental research has concentrated on using bioactive compounds to promote bone development locally. Several methods have been utilized in the literature, such as guided bone regeneration, alveolar osteodistraction, titanium meshes, and block/particle grafts. Other techniques for guided tissue regeneration and osseointegration are dental implant surface modifications and coating of the surface with osteoinductive biomaterials. All these materials have their specific indications with particular advantages and indications [3,4].

To overcome these problems, historically, autogenous bone grafts from the same patient’s iliac crest, mandible ramus, or chin have been used for alveolar reconstruction because of their osteoconductive and immunogenic qualities. Unfortunately, infection, discomfort, sensory loss, and donor site hematomas are common consequences of autogenous bone graft treatments. Moreover, a bone-rich donor site is not always available. Allograft bones, processed and controlled by a tissue bank or commercial source, are routinely used alternatives. This approach also has some drawbacks, including uneven osteoinductive efficacy, adverse host immunological responses, delayed resorption, and possible virus transmission [5,6].

The perfect implant dentistry bone biomaterial should have the following features: it should be biomimetic, induce endothelial and osteoblastic cell differentiation for new bone formation, have no immune-stimulating qualities, be readily manufactured or produced rather than taken from allograft materials (this is to prevent disease transmission), and be readily and rapidly reabsorbed as the osteogenic process takes place [7,8,9,10,11].

Growth factors (GFs) are proteins that are essential for tissue regeneration, development, and embryogenesis. Unlike certain hormones that control the growth of entire organisms, GFs are necessary for both the replication of individual cells and the preservation of normal cell function. While certain GFs are exclusive to specific cells, others promote cell division in a wide variety of cell types [12]. They attach to surface receptors to mediate their actions on cells. Signaling systems within cells are activated when GFs bind to their associated receptors. These signaling cascades control the transcription of genes involved in cellular functions, like proliferation, differentiation, migration, protein synthesis, and metabolism [1,5].

GFs can be of blood-borne and recombinant types. In the first kind, GFs are extracted from the cells of the patient. The most frequent source of GFs are platelets. Platelet-rich fibrin (PRF), platelet-rich plasma (PRP), and concentrated growth factor (CGF) are the primary subtypes of this harvest. Recombinant GFs were produced as mass manufacturing of any protein became possible [13]. Thanks to developments in recombinant technology, proteins can now be synthesized under regulated conditions, allowing for the mass manufacturing of concentrated and pure molecules. As a result, recombinant growth factor/matrix combination products have been developed and brought to the market. Combination products have drawn more attention as a means of maximizing tissue regeneration and constitute a new and developing paradigm in regenerative therapies [4,12].

The main procedure for producing GFs from blood is drawing blood, then centrifuging it to separate the plasma or serum. After that, plasma or serum is treated, usually at the filtration; concentration; and, occasionally, activation stages to separate and enrich particular GFs, such transforming growth factor beta (TGF-β), vascular endothelial growth factor (VEGF), or platelet-derived growth factor (PDGF). Preparing platelet-rich plasma (PRP) is a popular technique that involves centrifuging blood to concentrate platelets, which are subsequently activated to release GFs [14].

The incorporation of GFs in dental implantology, either as an osteoinductive material to improve implant site bone quality and quantity or as bioactive coating on the implant surface, offers a cutting-edge method in bone tissue engineering to establish ideal circumstances for bone healing, which has a significant impact on implant attachment [4,15]. The number of relevant studies has significantly risen since the discovery of several GFs that affect bone regeneration.

## 2. Aims

The aim of this narrative review is to provide an update for researchers and clinicians on the application of GFs in implant dentistry with a special emphasis on human clinical trials. However, since wound healing and osseointegration are pre-requisites of successful implantation, and, on the other hand, GFs are important components of the wound healing process, this review first starts with the basic biology of wound healing. Then, the role of GFs in wound healing and tissue regeneration will be presented. Finally, results of the published and relevant literature between 2000 and 2025 on different applications of GFs in the field of implant dentistry will be critically discussed. Different types of GFs, techniques of GF preparation from the blood, and application strategies are not addressed in this review, since there are plenty of high-quality papers that sufficiently cover this aspect.

## 3. Physiology of Wound Healing

Both soft- and hard-tissue healing are needed for a successful implant. No matter how biomimetic an implant is, there will always be an early injury and inflammation phase after the placement of dental implants. This section provides a short overview of the biological basis of wound healing.

The coordinated reaction to tissue damage includes inflammation and wound healing. To eradicate the injury, cellular and vascular inflammation is needed. Wound healing rebuilds damaged tissues to restore function. Thus, inflammation and wound healing are independent and well-defined humoral and innate responses to tissue damage that allow the tissue to recover [16].

A range of cells, including platelets, monocytes, lymphocytes, and polymorphonuclear leukocytes (neutrophils), infiltrate the injured tissues as part of the inflammatory response [17]. In early phases of inflammation, neutrophils outnumber macrophages. Macrophages regulate immunomodulation, phagocytosis, and antigen presentation and cause, maintain, and resolve inflammation. Platelets are essential to inflammation, especially following trauma [18]. Platelets are a major source of GFs, vasoactive amines, and several inflammatory mediators that serve as chemotactic signals for a variety of downstream inflammatory reactions, including phagocytosis, cell recruitment, and the synthesis of a new extracellular matrix [16]. Lymphocytes can be classified as B or T cells. Broadly speaking, T cells are involved in cell-mediated host responses, which are regulated by a variety of cytokines that interact with unique cell surface receptors, whereas B cells generate antibodies locally [17].

Inflammation is acute or chronic. After a damaging stimulus, the body reacts with acute inflammation. Leukocytes and plasma, mostly granulocytes, from the circulation enter wounded tissues, causing this response. Several biological pathways cause and propagate inflammation. Immune cells, local vascular cells, and damaged tissue cells participate in these processes. Chronic inflammation, on the other hand, involves tissue breakdown and healing and a constant change in cell types, such as mononuclear cells [16,19]. Acute inflammation ceases when leukocytes escape the lymphatic system or undergo apoptosis, starting wound healing. Chronic inflammatory disorders, like periodontitis, result from untreated acute inflammation. There are direct and indirect links between acute inflammation, resolution, chronic inflammation, and wound healing [20].

Wound healing involves hemostasis, inflammation, proliferation, and remodeling. Clotting factors restrict blood loss from the wound site and provide a foundation and structural matrix for granulation tissue during hemostasis, which occurs immediately after an injury. This happens shortly after an injury. In acute wounds, phagocytic cells that create reactive oxygen species can prolong the inflammatory phase for seven days. In chronic wounds, this phase may last longer [21,22]. The proliferative phase of wound healing begins when inflammatory cells die. This phase includes blood vessel production, granulation tissue, wound contraction, and epithelialization. The last remodeling phase, which involves scar tissue formation, can take months or years, depending on the wound’s location, severity, and treatment [23,24]. Figure 1 illustrates the phases of wound healing.

## 4. Role of Growth Factors in Tissue Regeneration, Healing, and Repair

GFs were originally characterized as secreted, physiologically active chemicals that affect cell proliferation. This term now includes secreted substances that affect cellular differentiation or mitosis. GFs can affect cell surface receptors, which deliver growth signals to intracellular components and modify gene expression. Signal transduction is the transfer of a chemical signal from outside to a cell to start a biological reaction [17,26].

GFs are usually proteins or peptides that have a significant affinity for a plasma membrane protein called a surface receptor. Peptides have two to fifty amino acid residues, while proteins have more than 50. Peptide/protein GFs bind to receptors on the outer cell membrane. Certain GFs are cytokines, which are tiny peptides [17,27]. Even though all cytokines affect signal transduction pathways, only those that affect cell growth/differentiation signaling pathways are GFs. GFs without surface receptors, such as lipid-soluble steroid hormones, can traverse the plasma membrane, connect to nuclear or intracellular protein receptors, and deliver a growth signal [14,28].

The essential components of wound healing and tissue regeneration include a good blood supply to the healing area, a bank of hard- and soft-tissue-forming cells, a scaffold and matrix that support the healing tissue, and growth and differentiation factors that lead the entire process [29,30]. Figure 2 illustrates these factors.

It is well known that GFs are essential for tissue growth and repair. Every repair stage is regulated by a variety of cytokines and GFs that act locally as regulators of basic cell functions through endocrine, paracrine, autocrine, and intracrine pathways. GFs regulate extracellular matrix protein synthesis and degradation, affecting angiogenesis, chemotaxis, and cell proliferation in tissue repair and illness [17,31]. They activate intracellular signal transduction cascades by binding to a target GF receptor’s extracellular domain. Since several of GFs’ roles in tissue repair have been clarified, controlled temporal expression is necessary following surgery [32]. In short, GFs can be seen in all the phases of wound healing. Table 1 summarizes the main growth factors and their roles in wound healing.

## 5. Application of Growth Factors in Implant Dentistry

In the last 30 years, growth factors have been used in different fields of dentistry, such as periodontics, oral medicine (particularly in the treatment of recurrent aphthous stomatitis), oral and maxillofacial reconstruction, the treatment of oroantral fistula, sinus elevation procedures, temporomandibular joint dysfunction, endodontics, guided bone regeneration (especially in the treatment of extraction sockets), alveolar osteitis, trismus, post-operative pain and swelling, the treatment of osteonecrosis, and dental implantology [33,34,35,36,37,38]. Despite this wide spectrum of applications, clinical research with this agent has mostly focused on peri-implant and periodontal regeneration purposes [4].

There are some published reviews on the role and utilization of GFs in dentistry [5,6,39,40]. However, the number of specific reviews on the utilization of GFs in implant dentistry is few. In 2010, Shimono et al. systematically reviewed the effect of GFs for bone augmentation to enable dental implant placement. They found varying degrees and amounts of evidence indicating that PRGF, rhPDGF, and rhBMP-2 may promote local bone augmentation under different circumstances; in particular, rhBMP-2’s potential appeared encouraging. However, they stated that the generalizability of this strategy was limited due to the small number of scientists employing these techniques and the small number of patient treatments documented in the literature [3]. In 2011, Kaigler et al. reviewed the role of PDGFs in periodontal and peri-implant bone regeneration. They concluded that rhPDGF-enhanced matrices can be used to promote periodontal and peri-implant bone regeneration [4]. In 2020, Lokwani et al. systematically reviewed the use of CGFs in implant dentistry. They stated that although more clinical studies are required to validate the potential merits of CGF in the long run, preliminary results seem promising, and CGF can promote osseointegration and enhance bone regeneration [12]. To the best of our knowledge, this is the first extensive review of solely clinical trials on the application of all types of GFs together with dental implant placement.

To investigate the main research question of this review, an electronic literature search was conducted in the PubMed database. PICOS/T components of the research question are presented in Table 2. The following search strategy was applied: (“growth factor” OR “platelet-rich fibrin” OR “PRF” OR “platelet-rich plasma” OR “PRP” OR “platelet-derived growth factor” OR “concentrated growth factor” OR “CGF”) AND “dental implant”. The following filters were utilized: clinical study, clinical trial, randomized clinical trial, and from 2000 to 15 May 2025. The initial search revealed 44 results. All the clinical human studies with the actual implant placement were included. All primary research types, regardless of the study design, were included, except case reports and case series with few cases. Despite the utilization of GFs, those studies that only investigated bone regeneration without the involvement of implants were excluded. The relevant literature between the years 2000 and 2025 were included. The selection of papers was based on the publication year, research methodology, research findings, and publication relevance. After applying the inclusion and exclusion criteria, a total of 29 records were selected to be included in the qualitative analysis of this review. No ethical approval was needed in the production of this review. Key characteristics of the included studies are presented in Table 3.

Most of the included studies in this review were published after 2015, indicating a rising trend in using GFs in dental implantology. The main clinical outcomes studied in the papers were as follows: implant survival, implant stability, alveolar bone gain, bone augmentation for sinus lifts, bone loss, ridge preservation after extraction, soft-tissue healing, and peri-implantitis.

Eleven included studies used PRF [44,51,53,54,55,57,61,62,63,65,66]. Recently, Anis et al. used PRF as a control group compared to a PRF + bone graft group, with no statistical difference between the two [53]. The fact that PRF was used as a control group may indicate the direction towards the establishment of PRF as a standard and predictable treatment. The results of these human clinical trials involving PRF and dental implants generally suggest that PRF can improve implant outcomes. According to these studies, PRF may increase implant stability, encourage bone regeneration surrounding the implant site, and accelerate soft-tissue recovery. Nevertheless, there was conflicting evidence, with some experiments showing no significant improvement when compared to controls. Overall, further high-caliber, extensive randomized controlled studies are required to draw firm conclusions about PRF’s effectiveness and the best application practices, even though it appears promising as a bioactive adjuvant to enhance healing in dental implant operations.

Three studies used PRP as their GF of choice [45,52,59]. ArRejaei et al. compared xenografts with PRP and reported significant differences [59]. Ntounis et al. stated that the use of PRP may enhance healing within extraction sockets and decrease the healing time prior to dental implant placement [52]. Shah et al. placed immediate implants pretreated with photofunctionalization (PF group) or PRP. Mean marginal bone loss was not significantly different in the PF group and the PRP group than the control group. On the other hand, the PF group and the PRP group showed significantly greater implant stability as compared to the control group [45]. It is difficult to draw a specific conclusion on PRP based on these few studies; however, it can be regarded as a safe supplement that could help dental implant treatments heal more quickly.

The results of studies that used CGF were heterogenous. For example, Sohn et al. demonstrated that on both conventional radiographs and cone-beam computed tomograms, new bone consolidation was seen along the implants in every patient. Following loading, the implant’s success rate was 98.2%. They suggested that as an alternative to bone grafting, the insertion of fibrin-rich blocks with CGFs demonstrated successful new bone development in the sinus, and this can be a predictable technique for sinus augmentation [47]. In addition, implant stability was documented to be significantly higher in implant cavities of a study group that were covered with CGF before implant placement [56]. Moreover, Chen et al. placed immediate implants after sinus floor elevation with CGF application. After six months, they found a significant gain in the alveolar bone height. They concluded that CGF application and simultaneous short implant placement could yield predictable clinical results for severely atrophic maxilla with a residual bone height [64]. In contrast, CGF did not contribute to implant stability in another study [67]. Isler et al. treated peri-implantitis by combining a bone substitute with collagen membrane (CM) or CGF. They proposed that both regenerating methods produced notable enhancements in radiographic and clinical evaluations [48]. The common point of these studies is the relatively small sample size of the implants. However, it can be inferred that CGF, either by itself or in conjunction with other common bone grafts, may help achieve vertical bone gain around implants. Additionally, using CGF may greatly enhance the quality of new bone that grows around implants. On the other hand, while the initial evidence appears encouraging, there are insufficient trials assessing the impact of CGF on soft-tissue healing, implant stability/survival, and sinus floor augmentation. Moreover, its antimicrobial effect and possible role in the treatment of peri-implantitis need to be studied in further investigations.

In an early and pioneering study in 2003, to enhance guided bone regeneration therapy regarding bone volume, density, and maturation, Jung et al. sought to determine whether adding recombinant human bone morphogenetic protein-2 (rhBMP-2) to a xenogenic bone graft would be beneficial. A resorbable collagen membrane and xenogenic bone substitute were added to both test and control defects, which were randomly assigned to test and control treatments. For both test and control locations, the decrease in defect height from the baseline to implant re-entry was statistically significant. According to this histomorphometric study, test sites had an average area density of 37% newly produced bone, while control sites had an area density of 30%. According to these findings, the xenogenic bone substitute and rhBMP-2 together can improve the maturation phase of bone regeneration and increase the graft to bone contact in patients. Guided bone regeneration therapy may be reliably enhanced and accelerated by rhBMP-2 [58].

Later, in 2005, Boyne et al. employed rhBMP-2 to safely and effectively induce sufficient bone for endosseous dental implants in patients who needed staged maxillary sinus floor augmentation. In this randomized controlled trial, the authors used a recombinant human protein to show de novo organ tissue growth in humans. They concluded that in patients who needed staged maxillary sinus floor augmentation, rhBMP-2/ACS safely produced enough bone for the implantation and functional loading of dental implants [46]. In contrast, in 2009, results of a multicenter, randomized, prospective clinical trial with 160 patients and 290 implants showed that the effectiveness of rhBMP-2/ACS is similar to the bone graft for sinus floor augmentation, although the outcome of functional loading was achieved [43].

GFs were used as coating materials in two studies [41,50]. In an investigation with a relatively large sample size of 241 patients and 1139 implants, the five-year survival of dental implants that were loaded immediately and bioactivated with plasma rich in GFs was investigated. For the implant-, surgery-, and patient-based analyses, the overall survival rates were 99.3%, 96.8%, and 96.9%, respectively. There was no significant correlation found between any of the examined variables and implant failures. The authors came to the conclusion that, when utilized in accordance with rigorous clinical guidelines, immediate loading of implants can be regarded as safe and predictable [41]. In a similar study, 61 implants coated with plasma rich in GFs were followed-up for one year, with an implant survival rate of 98.4% [50]. However, the focus of these studies was the immediate implant placement rather than the effect of the GF coating. Since none of these studies contained non-coated controls that could allow for a comparison, the actual impact of surface coating by GFs is still missing. Surface implant coating with various biomimetic substances may help increase osseointegration and decrease peri-implantitis. Therefore, GFs as surface coatings for implants deserve further inquiry and investigation. Figure 3 shows several materials that could be used as implant surface modifiers.

Amorfini et al. employed a corticocancellous allograft block or deproteinized bovine bone in conjunction with autologous bone, either alone or in combination with recombinant human platelet-derived growth factor-BB (rhPDGF-BB) in 50 implants. After a year of functional loading, the groups’ regenerated bone volume outcomes were comparable, but rhPDGF-BB had a beneficial effect on soft-tissue healing [49]. In a comparable study, rhPDGF-BB with hydroxyapatite was used and compared to an autogenous bone block. Neither the bone crest width nor implant torque were significantly different between the studied groups [60].

In a retrospective study, Mozzati et al. looked at the clinical records of 235 middle-aged women who received oral BP therapy for osteoporosis and had 1267 dental implants placed. The implants were placed with PRGF. They investigated bisphosphonates-related osteoporosis of the jaws (BRONJ) and implant failure as outcomes. At up to 120 months of follow-up, 16 patients lost 16 implants, resulting in a 98.7% implant-based survival rate and a 93.2% patient-based survival rate. There were no BRONJ cases reported. They concluded that patients receiving oral BPs continue to have a low chance of developing BRONJ because of dental implant surgery, especially when employing techniques like platelet concentrations that have the potential to promote and facilitate recovery [42]. However, it is worthy to note that the American Association of Maxillofacial Surgeons in their paper in 2022 still considered it prudent to recognize the risk of BRONJ development among patients on bisphosphonates [70].

This review has some limitations. The extensive variability in the exposure (GF types, modes of application, sample sizes, patient characteristics, follow-up durations, …etc.) and the heterogeneity of the outcomes (implant stability, implant survival, subjective pain sensation, patient satisfaction, bone gain, differences in the operational procedures, …etc.) made it difficult to draw uniform conclusions. We tried to overcome this limitation by grouping the exposures and the outcomes and apprising their effects. Another limitation is the inherent subjectivity of narrative reviews. To overcome this limitation, we applied a reproducible search strategy and distinct inclusion and exclusion criteria to make the selection and analysis process more systematic.

## 6. Conclusions

Numerous clinical investigations are currently being conducted on the application of GFs in dental implantology. The majority of the research supports the use of GFs, and several studies have shown encouraging findings. However, given the shortcomings of earlier studies, this topic requires more investigation. Standard GF application protocols do not yet exist. This could help to explain some reported mixed results and some inconsistencies between investigations. It is undeniable that the clinician’s skill and knowledge of the preparation technology are critical to the successful use of a GF product. There is a chance that GF will be used in oral implantology more frequently due to the increasing popularity of centrifuges and the ease of treatments at dental clinics.

Specifically, GFs can enhance surgical results in the operating field, such as increasing the height and thickness of the alveolar bone, and patients’ overall quality of life when compared to traditional techniques, i.e., without the use of GFs. Undoubtedly, further research is required to examine the variations among the different GFs and their effectiveness in greater depth.

PRF was the most widely used type of GF in the examined studies. PRF may increase implant stability, encourage bone regeneration surrounding the implant site, and accelerate soft-tissue recovery. Nevertheless, there is conflicting evidence, with some experiments showing no discernible improvement when compared to controls, even though it appears promising as a bioactive adjuvant to enhance healing in dental implant operations.

Concerning the CGF, it can be concluded that either by itself or in conjunction with conventional bone grafts, it may help achieve vertical bone gain surrounding implants. Additionally, using CGF greatly enhances the quality of new bone that grows around implants. On the other hand, while the initial evidence appears encouraging, there are insufficient studies assessing the impact of CGF on sinus floor augmentation, implant stability, implant longevity, and soft-tissue healing.

Few studies have been conducted on using GFs as a coating material to make implant surfaces more bioactive. This also applies to GFs’ antibacterial properties and their use in the management of peri-implantitis. Further research is required to examine each of these elements.

Understanding the cellular and molecular foundations of signaling pathways of bone regeneration and creating suitable carriers for GFs will undoubtedly spark a significant revolution in dentistry, enabling regenerative processes to take precedence over cicatricial ones. Nevertheless, there are still insufficiently few well-constructed randomized clinical trials to develop clinical guidelines for the utilization of GFs in the integration process of an implant or improving the recipient bone bed before its placement.

## Figures and Tables

**Figure 1 cimb-47-00317-f001:**
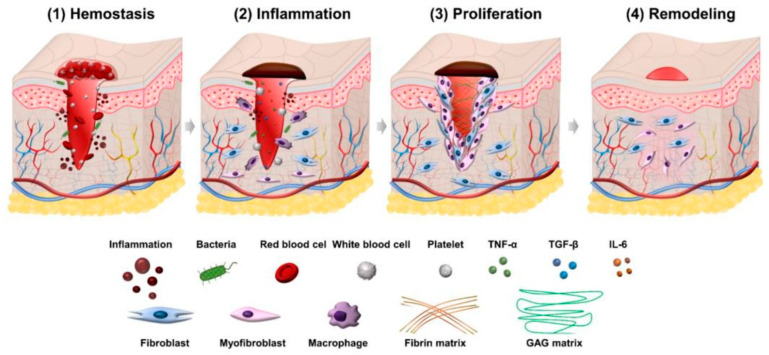
Stages of wound healing. Reproduced from [25]. GAG: glycosaminoglycan; IL: interleukin; TGF: transforming growth factor; TNF: tumor necrosis factor.

**Figure 2 cimb-47-00317-f002:**
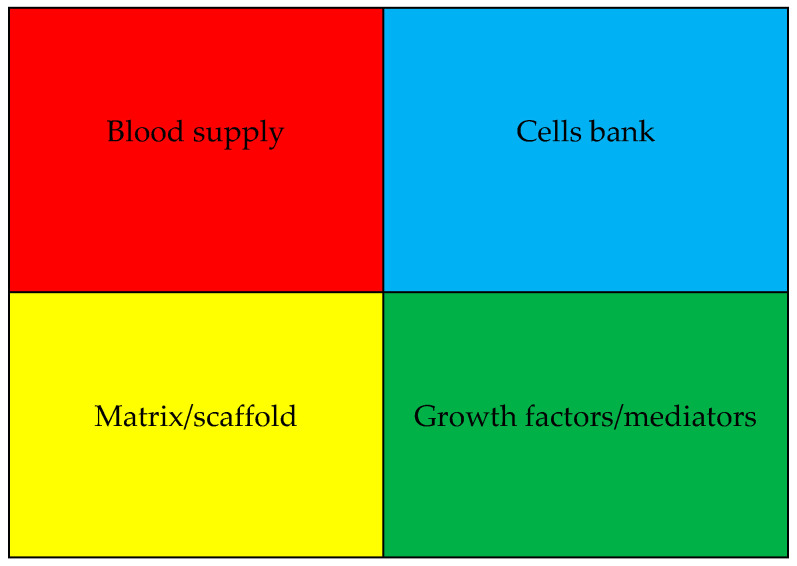
Basic elements required for wound healing, repair, and regeneration: sufficient blood flow, a source of cells to build soft- and hard-tissue structure, a scaffold or matrix for support, and growth factors that control cell migration, proliferation, synthesis, and angiogenesis for the site’s revascularization.

**Figure 3 cimb-47-00317-f003:**
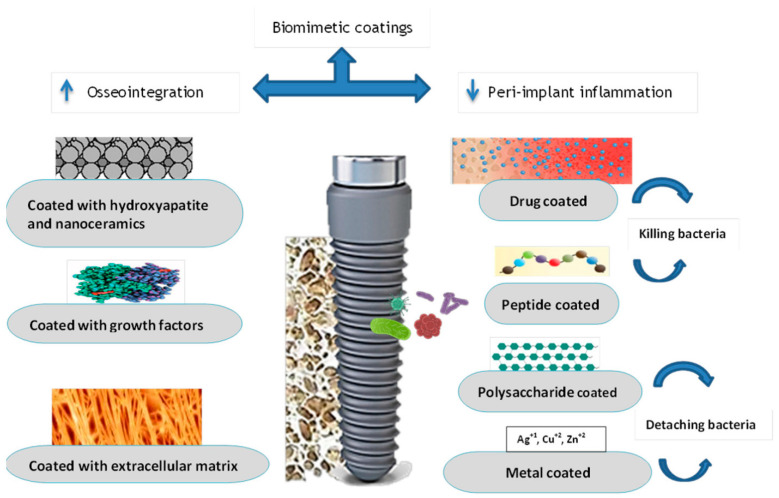
Biomimetic coatings of dental implants, including growth factors.

**Table 1 cimb-47-00317-t001:** Main growth factors and their roles in wound healing.

Wound Healing Phase	Growth Factor Type	Cell of Origin	Functions
INFLAMMATORY	PDGF	Platelets	Enhances neutrophil and monocyte chemotaxis.
	TGF-β	Platelets, leukocytes, and fibroblasts	Promotes neutrophil and monocyte chemotaxis. Additional cytokines (TNF-α, IL-1β, PDGF, and chemokines) are produced by autocrine expression.
	VEGF	Platelets, leukocytes, and fibroblasts	Increase vascular permeability.
PROLIFERATIVE	EGF	Macrophages, mesenchymal cells, and platelets	Promotes the migration and proliferation of epithelial cells.
	FGF-2	Macrophages and endothelial cells	Promotes the growth of fibroblasts and the production of extracellular matrix. Boost endothelial cell chemotaxis, proliferation, and differentiation.
	KGF(FGF-7)	Keratinocytes and fibroblasts	Promotes the migration and proliferation of epithelial cells.
	PDGF	Macrophages and endothelial cells	Promotes the growth of fibroblasts and the production of extracellular matrix. Boost endothelial cell chemotaxis, proliferation, and differentiation.
	TGF-β	Macrophages, fibroblasts, and leukocytes	Promotes the migration and proliferation of epithelial cells. Promotes the growth of fibroblasts and the production of extracellular matrix. Increases the synthesis of inhibitors and inhibits proteases.
	VEGF	Macrophages	Enhances endothelial progenitor cell chemotaxis. Promotes the growth of endothelial cells.
BONE REMODELLING AND MATRIX SYNTHESIS	BMPs 2–4	Osteoblasts	Promotes the migration of progenitor cells from mesenchyme.
	BMP-7	Osteoblasts	Enhances the differentiation of osteoblasts and chondroblasts.
	FGF-2	Macrophages and endothelial cells	Activates progenitor cell migration from the mesenchyme.
	IGF-2	Macrophages and fibroblasts	Promotes proliferation of osteoblasts and the synthesis of bone matrix.
	PDGF	Macrophages	Enhance differentiation of fibroblasts into myofibroblasts. Promotes the proliferation of progenitor cells from mesenchyme.
	TGF-β	Fibroblasts and osteoblasts	Prompts the apoptosis of endothelial cells and fibroblasts. Prompts the differentiation of fibroblasts to become myofibroblasts. Activates survival of osteoblasts and chemotaxis.
	VEGF	Macrophages	Chemotaxis of stem cells from the mesenchyme, antiapoptotic effect on bone-forming cells, and promotes angiogenesis.

BMP: bone morphgenetic protein; EGF: epidermal growth factor; FGF: fibroblast growth factor; IGF: insulin-like growth factor; TGF: transforming growth factor; PDGF: platelet-derived growth factor; KGF: keratinocyte growth factor; VEGF: vascular endothelial growth factor. Reproduced from [29].

**Table 2 cimb-47-00317-t002:** PICOS/T components of the main research question.

Sections	Description
Problem/ Population	Does the application of growth factors positively affect dental implant outcomes clinically? Complete and partially edentulous human patients receiving dental implants.
Intervention	Use of growth factors alone or with other bone graft materials.
Comparison	Intragroup comparison between baseline and after treatment or intergroup group comparison with the controls.
Outcome	All the reported clinical, radiological, and statistical outcomes.
Study design/ Time	All the primary research study designs except single case reports and series with few cases (<10 cases). All the published literature in the mentioned databases between 2000 and 2025.

**Table 3 cimb-47-00317-t003:** Summary characteristics of the included studies, listed according to the sample size.

Author (Year) [Reference]	Sample Size	Type of GF Used	Follow-Up Duration	Measured Outcomes	Findings
Anitua (2008) [41]	241 patients (1139 implants)	Implants with platelet-rich growth facor (PRGF)	5 years	Implant survival	The corresponding survival rates for the implant-, surgery-, and patient-based analyses were 99.3%, 96.8%, and 96.9%.
Mozzati (2015) [42]	235 female patients on bisphosphonates (BPs) (1267 implants)	Plasma rich in growth factor (PRGF)–Endoret	10 years	BP-related osteonecrosis of the jaws (BRONJ) and failure rate	Survival rate was 98.7% and 93.2% on implant basis and patient basis, respectively. No cases of BRONJ were reported.
Triplett (2009) [43]	160 patients (490 implants)	Human morphogenetic protein-2 (rhBMP-2) and collagen sponge (ACS) compared to autogenous bone graft	6 months	New bone formation, placement integration, and functional loading	Implants placed in rhBMP-2/ACS and bone graft groups performed similarly after functional loading.
Simonpieri (2017) [44]	42 patients (334 implants)	PRF in buccal bone augmentation	4 months	Implant survival and radiographic bone loss	Implants in the maxilla had 97.8% survival; the mandible, 98.1%; immediate implants, 98.3%; and delayed implants, 96.9%. There were no significant differences (*p* > 0.05) in mean radiographic bone loss between immediate and delayed implants or anterior and posterior implants.
Shah (2021) [45]	90 patients (90 implants)	Photofunctionalization (PF group) or platelet-rich plasma (PRP group)	1, 2, 4, 6, and 12 months	Biological outcomes (mean marginal bone loss and implant stability), esthetic outcomes (pink esthetic score and white esthetic score), and survival rate	PF and PRP groups had similar mean marginal loss compared to control group. The PF and PRP groups had considerably higher implant stability than the control group. Pink and white estheticscores were similar across groups.
Boyne (2005) [46]	48 patients (219 implants); 18 patients received 0.75 mg/mL of rhBMP-2/ACS 17 patients received 1.50 mg/mL of rhBMP-2/ACS; 13 patients (bone graft)	Group 1: 0.75 mg/mL of rhBMP-2/ACS; group 2: 1.50 mg/mL of rhBMP-2/ACS; group 3: standard bone graft material	6 months (bone density) and 36 months (implant survival and functionality)	Bone density and implant survival	For the bone graft, 0.75 mg/mL, and 1.50 mg/mL rhBMP-2/ACS therapy groups, the new bone density at 4 months post-operatively was 350 mg/cc, 84 mg/cc, and 134 mg/cc, respectively. These differences were statistically significant. Post functional survival at 36 months was 67%, 76%, and 62% in group 1, group 2, and group 3, respectively.
Sohn (2011) [47]	53 patients (113 implants)	Fibrin-rich blocks with concentrated growth factors (CGFs)	10 months	Bone formation and implant survival	On both conventional and cone-beam computed tomograms, new bone consolidation was seen along the implants in every instance. After loading, the implant’s success rate was 98.2%.
Isler (2018) [48]	52 patients with at least one peri-implantitis case	A bone graft combined with either collagen membrane (CM) or concentrated growth factor (CGF)	6–12 months	Bleeding on probing (BOP), gingival index (GI), clinical attachment level (CAL), probing depth (PD), and mucosal recession (MR)	Significant reductions were obtained in the studied parameters at both 6 and 12 months post-operatively for both treatments. The mean PD, CAL, and vertical defect depth (VDD) values were statistically significant in favor of the CM group at 12 months (*p* < 0.05), but at 6 months, no statistically significant difference was seen for any of the clinical parameters between the groups.
Amorfini (2013) [49]	16 patients (50 implants)	Autologous bone supplemented with bovine bone, either alone or in combination with recombinant human platelet-derived growth factor-BB (rhPDGF-BB)	12 months	Quantity of bone variation	The two groups’ changes in bone volume did not differ substantially (*p*-value = 0.25).
Fabbro (2009) [50]	30 patients (61 implants)	Implants coated with plasma rich in growth factors	1 year	Implant survival	Implant survival was 98.4% at 1 year of function.
Torkzaban (2018) [51]	10 patients (50 implants)	PRF group vs. no PRF group	Surgery day (T1), at 1 week (T2), and at 1 month (T3)	Implant stability measured by resonance frequency	After 1 week (T2), the PRF group had a mean implant stability quotient (ISQ) of 59.85 ± 5.32, while the non-PRF group was 55.99 ± 3.39. The ISQ rose to 0.12 ± 0.47 (*p* = 1.000) in the PRF group and fell to 2.42 ± 0.36 (*p* < 0.001) in the non-PRF group compared to the baseline. At 1 month post-op, ISQ significantly rose by 6.89 ± 0.96 in PRF and by 4.82 ± 0.92 in non-PRF compared to the baseline (*p* < 0.001).
Ntounis (2015) [52]	41 patients (40 implants)	Group 1, collagen plug (control); Group 2, FDBA/β-tricalcium phosphate (β-TCP)/collagen plug; Group 3, FDBA/β-TCP/platelet-rich plasma (PRP)/collagen plug; Group 4, FDBA/β-TCP/recombinant human platelet-derived growth factor BB (rhPDGF-BB)/collagen plug	After 8 weeks of healing, implants were placed	Subjective assessment of bone quality	Bone grafting changed D4 bone to D3 bone. PRP in bone grafting changed D4 bone, establishing D3 and D2 bones (56% vs. 42%). Combining rhPDGF-BB with β-TCP with bone grafting yields similar effects, but D2 quality is less common. Sockets with growth factors showed less remaining bone graft particles compared to those with FDBA/β-TCP/collagen plug alone.
Anis (2024) [53]	40 patients with atrophic maxilla treated by split-crest technique (40 implants)	Control group (PRF membrane) and test group (PRF membrane + Nanobone^®^)	5 months	Bone resorption and gain measured by CBCT	Horizontal bone width gain was 1.46 ± 0.44 mm for the control group and 1.29 ± 0.73 mm for the test group, with no statistical significance.
Diana (2018) [54]	29 patients (39 implants)	PRF group vs. non PRF group	3 months and 1 year	Implant stability	Both study and control groups showed significant increases in implant stability over 3 months (implant stability quotient: from 56.58 ± 18.81 to 71.32 ± 7.82; control group: from 60.61 ± 11.49 to 70.06 ± 8.96; *p* = 0.01). Implant stability was similar between groups.
Cheruvu (2023) [55]	40 patients with edentulous posterior mandibular sites	Group I received implants with a PRF membrane; group II was treated with implants alone	Baseline, and at 3-month and 6-month follow-ups.	Modified plaque index (mPI), gingival index (GI), width of keratinized tissue (WKT), thickness of keratinized tissue (TKT), and crestal bone level (CBL), assessed using digital intraoral periapical radiography (IOPA)	Significant increases in WKT and TKT were seen in both groups at 3 and 6 months post-op compared to the baseline (*p* < 0.05). Group I showed significant increases compared to group II (*p* < 0.05). Both groups showed significant increases in CBL at 3 and 6 months post-op (*p* < 0.05), with no distinguishing differences. CBL reduced in group I compared to group II at 3- and 6-month intervals (*p* < 0.05).
Pirpir (2017) [56]	12 patients (40 implants)	In contrast to the control group’s conventional implants, the study group’s implant cavities were coated with CGF prior to implant insertion	1 and 4 weeks	Implant stability quotient (ISQ)	By the first week, the study group’s mean ISQ value was 79.40, and the control group’s was 73.50; by the fourth week, they were 78.60 and 73.45, respectively. There were statistically significant differences between the groups (*p* < 0.05).
Kanayama (2016) [57]	27 patients (39 implants)	PRF as the only graft in sinus floor elevation. Two implant types were used: hydroxyapatite (HA) and sandblasted acid-etched (SA)	1 year	Bone gain	SA and HA groups had mean residual bone measures of 2.85 and 2.68 mm before surgery. The SA and HA groups had 4.38 and 4.00 mm mean annual bone increases.
Jung (2003) [58]	11 patients (34 implants)	Xenogenic bone graft and collagen membrane coated with rhBMP-2 (test); xenogenic bone graft and collagen membrane (control)	6 months	Bone volume, density, and maturation	The mean peri-implant bone defect was 5.8 mm in the control baseline and 7 mm in the test. The mean dropped to 0.02 for the test and 0.04 for the control at re-entry. There was statistical significance in this outcome (*p* < 0.01). According to a histological analysis, test sites had an average area density of 37% newly produced bone, while control sites had an area density of 30%.
ArRejaie (2016) [59]	16 patients (32 implants)	PRP gel plus bovine-derived xenograft vs. withour PRP	6 and 12 months	Dehiscence around immediate implants	Both treatments significantly improved the bone fill and marginal bone level. Statistically significant variations in bone density were seen between the control and the combined therapy (*p* ≤ 0.01).
Santana (2015) [60]	30 patients (30 implants)	rhPDGF with beta-tricalcium phosphate (β-TCP)/hydroxyapatite compared to autogenous bone block	6 months	Bone crest width (BCW) and implant torque	The experimental group’s mean baseline BCW reading was 3:03 mm, while the control group’s was 3:13 mm. In 87% of the experimental sites and 93% of the control sites, the implant was positioned with torque values greater than 35 N/cm. These two findings were not statistically significant.
Öncü (2019) [61]	26 patients (60 implants: 30 tests and 30 controls)	Test sockets were coated with leukocyte platelet-rich fibrin (L-PRF), and control sockets were not	1 week and 1 month	Implant stability and marginal bone loss	After 1 week and 1 month, the test group’s stability was higher (*p* < 0.002), and the group that received the PRF had a significantly smaller mean marginal bone resorption difference (*p* ≤ 0.05).
Singhal (2022) [62]	15 patients (30 implants)	Platelet-rich fibrin matrix (PRFM) with and without peripheral blood mesenchymal stem cells (PBMSCs)	1 week, 1 month, and 3 months	Implant stability	G1 and G2 insertion torque values were not significantly different (*p* = 0.81). PBMSCs and a platelet-rich fibrin matrix improved implant stability, with highly significant ISQ values at 1 week (*p* = 0.18), 1 month (*p* ≤ 0.001), and 3 months (*p* ≤ 0.001) in the G2 group.
Darestani (2023) [63]	14 patients (28 implants)	Eukocyte- and platelet-rich fibrin (L-PRF) vs. control	1, 2, 4, 6, 8, and 12 weeks	Implant stability	Both groups had significant stability results (*p* < 0.001; Eta2 = 0.322), although there was no significant difference in ISQ scores between the two groups (*p* > 0.05).
Chen (2016) [64]	16 patients (25 implants)	Immediate implants placed after sinus floor elevation with CGF application	19.8 months	Implant survival and vertical bone gain (VBG)	Implants had a 100% survival rate. Immediately following surgery, the mean VBG was 9.21 mm. The alveolar bone height (2.90 ± 0.22 mm) was significantly reduced 6 months later (*p* < 0.05). Additional alveolar bone resorption (0.14 ± 0.11 mm) was observed during the second 6-month period; however, it was not significant (*p* > 0.05).
Hartlev (2021) [65]	27 patients (27 implants)	Bone grafts covered by either a (PRF) membrane (PRF group) or coverage of the bone graft with deproteinised bovine bone mineral and a resorbable collagen membrane (control group)	2 years	Survival and marginal bone loss	The control group lost two implants (85% survival rate), while the PRF group lost none (100% survival rate). At follow-up, the PRF group had a mean marginal bone level of 0.26 mm (95% CI: 0.01–0.50 mm) and the control group, 0.68 mm (95% CI: 0.41–0.96 mm). A statistically significant difference between groups was −0.43 mm (95% CI: −0.80 to −0.05 mm; *p* = 0.03). At the final follow-up, both groups had healthy peri-implant soft tissue.
Amer (2024) [66]	24 patients (24 implants)	Autogenous demineralized dentin graft with injectable platelet-rich fibrin (ADDG + i-PRF) versus ADDG alone	6 months	Alveolar ridge preservation	ADDG alone or in combination with i-PRF produces similar clinical effects for ARP, osseous tissue quality, and patient satisfaction. However, adding i-PRF to ADDG preserves keratinized tissue and reduces post-operative discomfort.
Ozveri (2020) [67]	12 patients (24 implants)	CGF placed in implant sockets compared to conventional implants	1, 2, and 4 weeks	Implant stability	The study group’s mean ISQ was 67.00 ± 4.573, while the control group’s was 64.75 ± 5.065. There was no statistically significant difference.
Kim (2014) [68]	11 patients (16 implants)	Immediate implants placed with CGF	23.8 weeks	Bone gain measured by CBCT and implant survival	Survival was 100%. Average bone gain above the sinus floor was 8.23 ± 2.88 mm in the axial aspect of CBCT.
Palermo (2022) [69]	10 patients (20 implants: 10 tests and 10 controls)	10 implants coated with CGF compared to non-coated controls	6 months	Crestal bone level, probing depth, and bleeding	CGF-coated group showed better crestal levels and probing and less bleeding when compared with the controls that showed bone resorption problems.

GF: growth factor; CGF: concentrated growth factor; CBCT: cone-beam computed tomography; ISQs: implant stability quotients; VBG: vertical bone gain; L-PRF: platelet-rich fibrin; BCW: bone crest width; rhBMP: recombinant bone morphogenetic protein; rhPDGF: recombinant platelet-derived growth factor; CM: collagen membrane; GI: gingival index; BOP: bleeding on probing; PD: probing depth; CAL: clinical attachment level; MR: mucosal recession; PRP: platelet-rich plasma; PRGF: plasma rich in growth factors; VDD: vertical defect depth.

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
