# Peer review of "Applications of Growth Factors in Implant Dentistry"

_cimb, 2025, doi:10.3390/cimb47050317_

Round 1

Reviewer 1 Report

Comments and Suggestions for Authors

Dental implants have emerged as a promising solution for the restoration of the comfort and health of the stomatognathic system. The dental implants integration into the bone, bone grafts and surrounding soft tissue healing is enhanced with the use of concentrated growth factors (CGFs), proteins which regulate this complex processes. They play an important role in cell migration, cell proliferation, and angiogenesis in the tissue regeneration phase.

As stated by the authors "The aim of this scoping review is to provide an update for researchers and clinicians on the application of GFs in implant dentistry with a special emphasis on human clinical trials. However, since in one hand wound healing and osseointegration are pre-requisites of a successful implantation, and on the other hand GFs are important components of the wound healing process, this review first starts with the basic biology of wound healing. Then the role of GFs in wound healing and tissue regeneration will be presented. Finally, results of the published and relevant literature between 2000-2025 on different applications of GFs in the field of implant dentistry will be critically discussed. Different types of GFs, techniques of GF preparation from the blood and application strategies were not addressed in this review, since there are plenty of high-quality papers that sufficiently covered this aspect."

Although this is interesting topic there are several issues which needs to be addressed before publication.

The authors in the aim mentioned that this is a scoping review. Having this in mind, the structure of the manuscript is not adequate. Complete sections are missing including Material and methods and Results.

Furthermore, it is commendable to include and remind the readers about the role of GF in the basic biology of wound healing. However, the sections Physiology of wound healing and Role of growth factors in tissue regeneration, healing and repair are too general and vague. The whole sections needs to be rewritten.

In the section Application of growth factors in implant dentistry the attempt was made to summarize the data regarding the GF and implant dentistry. Some information regarding the methodology are presented. However, this is not enough and it is not adequately presented.

The PICO (Patient, Intervention, Comparison, Outcome) framework which is commonly used to develop focused clinical questions for quantitative systematic reviews was used. Although this is acceptable, please explain why not use the PRISMA Extension for Scoping Reviews as the reporting guideline for the scoping review manuscript.

What was the review protocol? What were the eligibility criteria? Provide rationale for all criteria.

Provide Information Sources and Search Strategy. Specify who develop search strategy. List all sources searched and general search terms used. For each source, include dates of search, limits/filters, general search terms. List software that will be used for citation management. Provide full search strategy for at least one database.

Provide Study Selection/Screening information. Describe title and abstract screening process, number of independent reviewers, how discrepancies were resolved, how unclear information was handled, and what software was used. Describe full-text screening process.

Provide Data Charting/Collection/Extraction information.

Provide Synthesis and Presentation of Results.

State whether ethics approval was required.

In Discussion section the extracted characteristics of the studies were not discussed. Discuss the importance of sample size, type of used GFs, follow-up duration, outcomes,… What is the role of GF in different outcomes.

Present the limitations of this scoping review.

You stated that "different types of GFs, techniques of GF preparation from the blood and application strategies were not addressed in this review, since there are plenty of high-quality papers that sufficiently covered this aspect". Maybe this is true, however you need to briefly address this issues in the present review.

Finally, what is the clinical significance? What is the need for this review, when as you mentioned here are plenty of high-quality papers that sufficiently covered similar aspects? The aim was provide an update for researchers and clinicians on the application of GFs in implant dentistry with a special emphasis on human clinical trials. When was the last scoping review published? Did it include animal and human studies?

Author Response

Dear and respected reviewer,

We are very glad to see that our manuscript has been thoroughly read, and valuable comments have been reported that definitely contribute to the development of the manuscript. Therefore, regardless of the fate of the manuscript, we express our sincerest thanks and gratitude for the time and effort given to review this manuscript.

The following are our point-to-point changes and responses:

Comment 1: [The authors in the aim mentioned that this is a scoping review. Having this in mind, the structure of the manuscript is not adequate. Complete sections are missing including Material and methods and Results.]

Response 1: [There has been a mistake from our side while naming type of the review, which is narrative and definitely not scoping. We are very sorry for this inconvenience. All the terms of “scoping” were changed. The structure of the manuscript is now compatible with a narrative review. After this change, an extra of 10 articles were added to the review that increased the number of included articles to 29]

Comment 2: [Furthermore, it is commendable to include and remind the readers about the role of GF in the basic biology of wound healing. However, the sections Physiology of wound healing and Role of growth factors in tissue regeneration, healing and repair are too general and vague. The whole sections needs to be rewritten.]

Response 2: [The two sections are rewritten and shortened. They are now less than 1000 words. The reasons for the presence of these two sections are the following: 1) The manuscript is submitted to the journal of Current Issues in Molecular Biology and a section called Bioorganic Chemistry and Medicinal Chemistry. There should be sufficient and multi-disciplinary background that fits the journal and the section. 2) We think background information on the usage of GFs in implant dentistry is necessary and rationalizes the whole application. 3) Also, it presents the biological links between the exposure (GF) and the outcome (dental implantology).

Comment 3: [In the section Application of growth factors in implant dentistry the attempt was made to summarize the data regarding the GF and implant dentistry. Some information regarding the methodology are presented. However, this is not enough and it is not adequately presented.]

Response 3: [Since we changed the format to narrative review and we included an extra of 10 ten articles, this section is rewritten. The changed or added parts are highlighted in bold in the manuscript.]

Comment 4: [The PICO (Patient, Intervention, Comparison, Outcome) framework which is commonly used to develop focused clinical questions for quantitative systematic reviews was used. Although this is acceptable, please explain why not use the PRISMA Extension for Scoping Reviews as the reporting guideline for the scoping review manuscript.]

Response 4: [Again, due to the change in the format, we think now the PICO serves the purpose.]

Comment 5: [What was the review protocol? What were the eligibility criteria? Provide rationale for all criteria.

Provide Information Sources and Search Strategy. Specify who develop search strategy. List all sources searched and general search terms used. For each source, include dates of search, limits/filters, general search terms. List software that will be used for citation management. Provide full search strategy for at least one database.

Provide Study Selection/Screening information. Describe title and abstract screening process, number of independent reviewers, how discrepancies were resolved, how unclear information was handled, and what software was used. Describe full-text screening process.

Provide Data Charting/Collection/Extraction information.]

Response 5: [All the previous comments are reasonable requests that a scoping review should contain. Although a narrative review does not have to contain a strict search strategy, screening and extraction information, we reported this in the following paragraph with the inclusion and exclusion criteria. This gave the review a more systematic approach and also reflected in the number of included studies:

To investigate the main research question of this review, an electronic literature search was performed in the PubMed database. PICOS/T components of the research question are presented in Table 2. The following search strategy was applied: The following search strategy was applied: ("growth factor" OR "platelet-rich fibrin" OR "PRF" OR "platelet-rich plasma" OR "PRP" OR "platelet-derived growth factor" OR "concentrated growth factor" OR "CGF") AND "dental implant". The following filters were utilized: clinical study, clinical trial, randomized clinical trial and from 2000 - 2025/5/15. The initial search revealed 44 results. All the clinical human studies with the actual implant placement were included. All primary research types regardless of the study design were included, except case reports and case series with few cases. Despite the utilization of GFs, those studies that only investigated bone regeneration without the involvement of implants, were excluded. Relevant literature between the years 2000 and 2025 were included. The selection of papers was based on the publication year, research methodology, research findings, and publication relevance. After applying the inclusion and exclusion criteria, a total of 29 records were selected to be included in the qualitative analysis of this review.]

Comment 6: [Provide Synthesis and Presentation of Results.]

Response 6: [Although this comment might be again related with the structure of a scoping review, we added the following paragraph for the presentations of the main results:

Most of the included studies in this review were published after 2015, indicating the rising trends of using GFs in dental implantology. The main clinical outcomes studied in the papers were: implant survival, implant stability, alveolar bone gain, bone augmentation for sinus lift, bone loss, ridge preservation after extraction, soft tissue healing and peri-implantitis.]

Comment 7: [State whether ethics approval was required.]

Response 7: [The following sentence is added: No ethical approval was needed in the production of this review.]

Comment 8: [In Discussion section the extracted characteristics of the studies were not discussed. Discuss the importance of sample size, type of used GFs, follow-up duration, outcomes,… What is the role of GF in different outcomes.]

Response 8: [Most parts of this section is rewritten/changed and some new paragraphs are added. The changed/added paragraphs are highlighted in bold in the manuscript.]

Comment 9: [Present the limitations of this scoping review.]

Response 9: [The limitation paragraph is added as the following:

This review had some limitations. The extensive variability of the exposure (GF types, mode of application, sample size, patient characteristics, follow-up duration…etc.) and the heterogeneity of the outcome (implant stability, implant survival, subjective pain sensation, patient satisfaction, bone gain, differences in the operational procedures …etc.) made it difficult to draw uniform conclusions. We tried to overcome this limitation by grouping the exposures and the outcomes and apprising their effects. Another limitation was the inherent subjectivity of narrative reviews. To overcome this limitation, we applied a reproducible search strategy and distinct inclusion and exclusion criteria to make selection and analysis more systematic.]

Comment 10: [You stated that "different types of GFs, techniques of GF preparation from the blood and application strategies were not addressed in this review, since there are plenty of high-quality papers that sufficiently covered this aspect". Maybe this is true, however you need to briefly address this issues in the present review.]

Response 10: [The following paragraph is added to the Introduction to address this point:

The main procedure for producing GFs from blood is drawing blood, then centrifuging it to separate the plasma or serum. After that, plasma or serum is treated, usually by filtration, concentration, and occasionally activation stages, to separate and enrich particular GFs such transforming growth factor-beta (TGF-β), vascular endo-thelial growth factor (VEGF), or platelet-derived growth factor (PDGF). Preparing platelet-rich plasma (PRP) is a popular technique that involves centrifuging blood to concentrate platelets, which are subsequently activated to release GFs.]

Comment 11: [Finally, what is the clinical significance? What is the need for this review, when as you mentioned here are plenty of high-quality papers that sufficiently covered similar aspects? The aim was provide an update for researchers and clinicians on the application of GFs in implant dentistry with a special emphasis on human clinical trials. When was the last scoping review published? Did it include animal and human studies?]

Response 11: [Thanks especially for this suggestion. We added this paragraph that mentioned the previous reviews and the originality of the present narrative review:

There are some published reviews on the role and utilization of GFs in dentistry [5,6,39,40]. However, the number of specific reviews on the utilization of GFs in im-plant dentistry is few. In 2010, Shimono et al., systematically reviewed the effect of GFs for bone augmentation to enable dental implant placement. They found varying degrees and amounts of evidence, indicating that PRGF, rhPDGF, and rhBMP-2 may promote local bone augmentation under different circumstances; in particular, rhBMP-2's potential appeared encouraging. However, they stated that the generalizability of this strategy was limited due to the small number of scientists employing these techniques and the small number of patient treatments documented in the literature [3]. In 2011, Kaigler et al., reviewed the role of PDGFs in periodontal and peri-implant bone regeneration. They concluded that rhPDGF-enhanced matrices can be used to promote periodontal and peri-implant bone regeneration [4]. In 2020, Lokwani et al., systematically reviewed the use of CGFs in implant dentistry. They stated that although more clinical studies are required to validate the potential merits of CGF in the long run, the preliminary results seem promising and CGF can promote osseointegration and enhance bone regeneration [12]. To the best of our knowledge, this is the first extensive review of the solely clinical trials on the application of all types of GFs together with dental implant placement.]

Reviewer 2 Report

Comments and Suggestions for Authors

Abstract It would be good to add what are the conclusions from your literature review research. What information is practical?
How many articles were reviewed, what were the criteria for acceptance or rejection?

Introduction
Dental implants, the most advanced - why the most advanced? Will you take up the evidence?
What thesis will you put at the beginning of your research?

What are the missing criteria for acceptance of articles and rejection? How many articles in total were found in the years 2000-2025?

Figure 1. GAG - please expand this abbreviation, provide captions under the figure explaining all the skins placed on it

Table 3 individual columns merge with each other, it is illegible

Is Figure 3 yours or was it taken from some source?

Conclusion

What are the limitations of your research?

Good luck with your future research!

Author Response

Dear and respected reviewer,

We are very glad to see that our manuscript has been thoroughly read, and valuable comments have been reported that definitely contribute to the development of the manuscript. Therefore, regardless of the fate of the manuscript, we express our sincerest thanks and gratitude for the time and effort given to review this manuscript.

The following are our point-to-point changes and responses:

Comment 1: [Abstract It would be good to add what are the conclusions from your literature review research. What information is practical?

How many articles were reviewed, what were the criteria for acceptance or rejection?]

Response 1: [Thanks for the suggestion. Due to word count limitations, we could only add the following short but precise sentence:

The PubMed database was searched using relevant keywords with some filters related to the research question. Out of the initial 44 records, all the clinical human studies (n=29) with the actual dental implant placement and its assessment, were included.]

Introduction

Comment 2: [Dental implants, the most advanced - why the most advanced? Will you take up the evidence?

What thesis will you put at the beginning of your research?]

Response 2: [Thanks again for the meticulous reading. The sentence is changed and now it is read as the following: Dental implants aiming to restore normal function and esthetics of missing teeth, are widely employed.]

Comment 3: [What are the missing criteria for acceptance of articles and rejection? How many articles in total were found in the years 2000-2025?]

Response 3: [The following highlighted part is added:

The following search strategy was applied: ("growth factor" OR "platelet-rich fibrin" OR "PRF" OR "platelet-rich plasma" OR "PRP" OR "platelet-derived growth factor" OR "concentrated growth factor" OR "CGF") AND "dental implant". The following filters were utilized: clinical study, clinical trial, randomized clinical trial and from 2000 - 2025/5/15. The initial search revealed 44 results. All the clinical human studies with the actual implant placement were included. All primary research types regardless of the study design were included, except case reports and case series with few cases. Despite the utilization of GFs, those studies that only investigated bone regeneration without the involvement of implants, were excluded. Relevant literature between the years 2000 and 2025 were included. The selection of papers was based on the publication year, research methodology, research findings, and publication relevance. After applying the inclusion and exclusion criteria, a total of 29 records were selected to be included in the qualitative analysis of this review.]

Comment 4: [Figure 1. GAG - please expand this abbreviation, provide captions under the figure explaining all the skins placed on it]

Response 4: [Explanation of the abbreviations are added]

Comment 5: [Table 3 individual columns merge with each other, it is illegible]

Response 5: [Thanks for the suggestion. Since tables the actual published papers are produced in landscape view, the columns will have more space the content can be read by the reader easily.]

Comment 6 [Is Figure 3 yours or was it taken from some source?]

Response 6: [Yes, it is from one of our publications.]

Comment 7: [What are the limitations of your research?]

Response 7: [The following limitation paragraph is added:

This review had some limitations. The extensive variability of the exposure (GF types, mode of application, sample size, patient characteristics, follow-up duration…etc.) and the heterogeneity of the outcome (implant stability, implant survival, subjective pain sensation, patient satisfaction, bone gain, differences in the operational procedures …etc.) made it difficult to draw uniform conclusions. We tried to overcome this limitation by grouping the exposures and the outcomes and apprising their effects. Another limitation was the inherent subjectivity of narrative reviews. To overcome this limitation, we applied a reproducible search strategy and distinct inclusion and exclusion criteria to make the selection and analysis more systematic.]

Comment 8: [Good luck with your future research!]

Response 8: [Thanks for your kind wish and we hope the same for you!]

Round 2

Reviewer 1 Report

Comments and Suggestions for Authors

The manuscript has been significantly improved with the made changes. The structure of the manuscript now fully corresponds to the narrative review. Please, in the future have in mind to check the type of the manuscript that you are submitting in order to avoid unnecessary inconvenience for the reviewers and you as the authors. Kind regards